# Mild to Moderate Iodine Deficiency and Inadequate Iodine Intake in Lactating Women in the Inland Area of Norway

**DOI:** 10.3390/nu12030630

**Published:** 2020-02-27

**Authors:** Synne Groufh-Jacobsen, Lise Mette Mosand, Ingvild Oma, Kjersti Sletten Bakken, Beate Stokke Solvik, Elin Lovise Folven Gjengedal, Anne Lise Brantsæter, Tor Arne Strand, Sigrun Henjum

**Affiliations:** 1Department of Research, Innlandet Hospital Trust, 2629 Lillehammer, Norway; tors@me.com; 2Department of Nursing and Health Promotion, Faculty of Health Sciences, OsloMet–Oslo Metropolitan University, 0130 Oslo, Norway; shenjum@oslomet.no; 3Department of Clinical Medicine, University of Oslo, 0450 Oslo, Norway; lise.mette.mosand@sykehuset-innlandet.no; 4Department of Medical Microbiology, Innlandet Hospital Trust, 2629 Lillehammer, Norway; ingvild.oma@sykehuset-innlandet.no; 5Women’s Clinic at Lillehammer Hospital, Innlandet Hospital Trust, 2629 Lillehammer, Norway; Kjersti.sletten.bakken@sykehuset-innlandet.no (K.S.B.); beate.stokke.solvik@sykehuset-innlandet.no (B.S.S.); 6Faculty of Environmental Sciences and Natural Resource Management, Norwegian University of Life Sciences, 1432 Aas, Norway; elin.gjengedal@nmbu.no; 7Division of Infection Control and Environmental Health, Norwegian Institute of Public Health, 0213 Oslo, Norway; annelise.brantsaeter@fhi.no

**Keywords:** Iodine, lactating women, Norway, breastfeeding, infants, iodine knowledge, human milk iodine concentration, iodine intake, iodine status, urinary iodine concentration

## Abstract

Breastfed infants are dependent on an adequate supply of iodine in human milk for the production of thyroid hormones, necessary for development of the brain. Despite the importance of iodine for infant health, data on Norwegian lactating women are scarce. We measured iodine intake and evaluated iodine status and iodine knowledge among lactating women. From October to December 2018, 133 mother–infant pairs were recruited in a cross-sectional study through two public health care centers in Lillehammer and Gjøvik. Each of the women provided two human milk specimens, which were pooled, and one urine sample for analysis of iodine concentration. We used 24-h dietary recall and food frequency questionnaire (FFQ) to estimate short-term and habitual iodine intake from food and supplements. The median (P25, P75) human milk iodine concentration (HMIC) was 71 (45, 127) µg/L—of which, 66% had HMIC <100 µg/L. The median (P25, P75) urinary iodine concentration (UIC) was 80 µg/L (52, 141). The mean (± SD) 24-h iodine intake and habitual intake was 78 ± 79 µg/day and 75 ± 73 µg/day, respectively. In conclusion, this study confirms inadequate iodine intake and insufficient iodine status among lactating women in the inland area of Norway and medium knowledge awareness about iodine.

## 1. Introduction

Adequate human milk iodine concentration (HMIC) is essential to provide sufficient iodine supply for the infant. Sufficient iodine intake during infancy is important to ensure optimal thyroid hormone stores and to prevent impaired neurological development [1,2,3,4,5]. The consequences of iodine deficiency depend on time of occurrence in life and its severity [6,7]. Early pregnancy is a particularly crucial period for fetal brain development. Adequate iodine supply throughout pregnancy, lactation and the first years of life is a prerequisite for normal neurodevelopment [5,6,7]. In 2018, estimates suggested that up to 50% of newborns in Europe were exposed to iodine deficiency [8], and hence at risk of not achieving their full intellectual capacity. 

Norway has been considered to be an iodine-sufficient country for decades, but recent evidence suggests that mild to moderate iodine deficiency is prevalent and increasing in some population groups, such as women of childbearing age [9,10,11,12]. The breastfeeding prevalence in Norway is generally high. According to the national breastfeeding report in 2013, 81% were still breastfed by the age of four months, 67% by the age of seven months, 55% by the age of 9 months, and 35% by the age of 12 months [13]. Due to high breastfeeding prevalence in Norway, breastfed infants are of particular concern. Lately, studies in Norway have found maternal iodine status to be associated with symptoms of child language delay, reduced fine motor skills and behavior problems at 3 years of age [14]. Furthermore, maternal iodine status has been associated with reduced expressive and receptive language [15], and possible effects in the child neurocognitive development at 8 years of age [16].

During lactation, nutritional requirements are increased, and the diet needs to ensure sufficient iodine status for both the mother and the infant. The Nordic Nutrition Recommendation (NNR) recommends an additional 50 µg/day (200 µg/day) during lactation to provide sufficient iodine in human milk [17]. After the 1950s, dairy products became the main source of iodine in Norway, especially milk and whey cheese, due to commissioning of mandatory iodization of the cow fodder to improve animal health. Lean fish, seafood and seaweed are also rich in iodine because marine plants and animals concentrate iodine from the seawater [6]. In Norway, there has been a decline in the intake of dairy products over the past decade, especially milk, and in the intake of lean fish [18]. Furthermore, the fortification of table salt is voluntary in Norway, and in insignificant amounts with only 5 µg/L. Therefore, the iodine contribution from table salt is considered negligible for the total iodine intake. A salt intake of 3 g a day only contributes with 15 µg of iodine [19]. For other food items (e.g., bread, bread products and cereals, vegetables (all types), potatoes, meat and meat products, fruits and berries (all types), fats and oils), the mean iodine concentration is assumed to be 2–3 µg/100gram in Norway [19,20]. Historically, the inland area of Norway had major iodine deficiency 100 years ago, with endemic goiter in up to 80% of all school children [21,22]. However, iodine status has not been measured in this area after the iodization of the cow fodder became mandatory. Considering the importance of sufficient HMIC and adequate maternal iodine intake during lactation, this study aimed to assess and evaluate iodine status and iodine knowledge in lactating women resident in the inland area of Norway.

## 2. Materials and Methods 

### 2.1. Study Population 

From October to December 2018, 133 mother–infant pairs were recruited through the public health centers in Lillehammer and Gjøvik (Figure 1). Healthy mothers who could read and write Norwegian and had a healthy infant between 0 and 12 months at study start were invited to participate. The inclusion criteria were: (1) No acute illness of the infant; (2) No known metabolic or congenital chronically illness of the infant, which is known to affect cognitive development; (3) Possibility to collect urine from the diaper and availability of blood sample of the infant. 

### 2.2. Study Design

### 2.3. Etichs

Prior to study start, all of the women provided written consent and the study had clearance from the Regional Committee for Medical and Health Research Ethics, 2018/1230/REC South East.

### 2.4. Collection of Human Milk and Urine Samples 

Two non-fasting human milk specimens per women were obtained in the same container (50 mL polypropylene (pp) centrifuge tube, Sarstedt, Nümbrecht, Germany)—one non-fasting spot sample (2.5 mL) in the morning before breastfeeding and one non-fasting spot sample (2.5 mL) in the afternoon after breastfeeding. The mothers were also instructed to collect non-fasting spot urine sample in a labeled 100 mL Vacuette^®^ urine breaker (Greiner Bio-One, Kremsmünster, Austria), and to withdraw a subsample of urine from the breaker into a 9.5 mL Vacuette^®^ Urine tube (Greiner Bio-One, Kremsmünster, Austria). Detailed oral and written information on how to collect and store the human milk and urine samples was given. The women recorded the day and time of sampling and whether they had eaten breakfast before donating the samples. In between sampling, the human milk and urine samples were stored in a refrigerator at 2–4 °C. Thereafter, all samples were stored at the Innlandet Hospital Trust in a freezer at −70 °C until analysis. 

### 2.5. Chemical Analysis

The determination of HMIC and UIC was performed at the Norwegian University of Life Science, Faculty of Environmental Science and Natural Resource Management. The human milk was thawed and then heated to 37 °C in a heating cabinet before being thoroughly homogenized. A conformance test between volume and weight of human milk limited iodine concentration to two significant figures. An aliquot of 1.00 mL of human milk was transferred into 15 mL pp centrifuge tubes (Sarstedt, Nümbrecht, Germany) by means of 100–5000 µL electronic pipette (Biohit, Helsinki, Finland) and diluted to 10.0 mL with an alkaline solution (BENT), containing 4% (weight (*w)*/volume(*V*) 1-Butanol and 0.1% (*w*/*V*) H_4_EDTA, 5% (*w*/*V*) NH_4_OH, and 0.1% (*w*/*V*) Triton^TM^ X-100. The urine was thawed but not heated before the alkaline dilution. In addition, the concentration of NH_4_OH in BENT was set to 2% to avoid precipitation of struvite (MgNH_4_PO_4_ ·6H_2_O) in the urine. Otherwise the sample preparation was equal to the one used for milk. Method blank samples and samples of standard reference material (SRM) were prepared in the same manner as the respective sample matrices. Reagent of analytical grade or better and deionized water (> 18 MΩ) were used throughout. The samples were analyzed for iodine concentration using an Agilent 8900 ICP-QQQ (Triple Quadruple Inductively Coupled Plasma Mass Spectrometer, Agilent Technologies, Hachioji, Japan) using oxygen reaction mode. Iodine was quantified on mass 127. In order to correct for non-spectral interference, we measured iodine-129 (^129^I).

For human milk, the limit of detection (LOD) and the limit of quantification (LOQ) were 0.04 and 0.14 µg/L, respectively. For urine, the LOD was 0.1 µg/L and the LOQ was 0.32 µg/L. The LOD and the LOQ were calculated by multiplying the standard deviation of the method blank samples (*n* = 5) by three and ten, respectively. To check for accuracy in the determinations of iodine in human milk and urine, SRM was used. Allowing for a coverage factor of k = 2—that is, a level of confidence of approximately 95%—our data were within the recommended values issued for the European Reference Materials (ERM)^®^-BD 150 and ERM^®^-BD 151 Skimmed milk powders and Seronorm^TM^ Trace Elements Urine L-1, Seronorm^TM^ Trace Elements Urine L-2. The measure repeatability was 1.6% with respect to both human milk and urine.

### 2.6. Iodine Intake From Food and Supplements

The food frequency questionnaire (FFQ) (habitual intake) had questions with seven alternative options, ranging from never/rarely to 5 or more times daily. Values for iodine concentrations in food were obtained from the 2018 version of the Norwegian Food Composition Table [23]. The daily consumption frequencies for all food items in the FFQ were multiplied with portion sizes and the iodine concentrations for each food item/dish, e.g., cow’s milk once daily with 2 dl per portion, with an iodine content of 16 µg per 100 g, is equal to 32 µg (16 µg × 2). For questions on food representing aggregated items with different iodine values, such as nuts (almonds, walnuts, etc.), lean fish (cod, saithe, etc.), sushi and cake, an average value for the food was calculated as reported in newly published studies on the iodine content in Norwegian foods [23,24]. When calculating the iodine intake by the 24-h recall, we multiplied food intakes with the iodine concentration in each food item and summed for all items. Because the 24-h recall only assessed iodine-rich foods, we added 15 µg of iodine to each individual 24-h recall to account for iodine from other sources. This fixed amount was based on the calculated iodine intake from other foods assessed by the mother’s habitual intake as used in previous studies [25,26].

Supplement consumption was assessed both by habitual and 24-h intakes. The participants reported frequency in times per week for each supplement consumed e.g., if a supplement contained 150 µg, and if taken 3 times a week, the contribution was estimated to be (150 µg × 3/7) 64 µg/day. 

### 2.7. Assessment of Iodine Knowledge

Iodine knowledge was assessed by using a previous validated questionnaire with six questions [26,27]. 

Three questions (What are the most important dietary sources of iodine?; Why is iodine important?; and What do you know about the current iodine status in Norway?) were used to calculate an iodine knowledge score. The three questions had multiple answer alternatives (some correct and some incorrect). Correct answers generated 2 points, correctly identified false answers generated 1 point, and incorrect answer gave 0 point. Participants who answered “don’t know” did not get 1 point, even if they “correctly identify false answer”. The total knowledge score ranged from 0 to 26 points and was further divided into four categories: poor knowledge score (0–5 points), low knowledge score (6–11 points), medium knowledge score (12–19 points) and high knowledge score (20–26 points). 

### 2.8. Definitions

Body Mass Index (BMI) was calculated as weight/height^2^ (kg/m^2^), with self-reported maternal weight and height before and after pregnancy. The World Health Organization (WHO)’s epidemiological criteria for assessment of iodine status in lactating women was used for evaluation UIC (≥ 100 µg/L) [28,29], but no exact cut-off value for the sufficient concentration of iodine in human milk is established. In this study, a median HMIC ≥ 100 µg/L at group level was considered as sufficient [28,30]. 

### 2.9. Statistical Analysis

IBM SPSS statistics version 24 and version 25 (IBM Corp., Armonk, NY, USA) were used for statistical analysis. For the construction of variables and figures, IBM SPSS (version 25) was used. The normality of the data was checked using Q–Q plots, histogram, and the Shapiro–Wilk test. To evaluate the agreement between continuous variables, we used Spearman’s correlation (r_s_) for non-parametric variables and Pearson’s correlation (r_p_) for normally distributed variables. We interpreted the strength of the correlation coefficients according to Cohen, 1988, and considered correlation (r) < 0.3 as weak, 0.3 ≥ r < 0.5 as moderate and r ≥ 0.5 as strong in this context. Multiple linear regression analyses were used to explore predictors of the dependent variable HMIC. A stepwise procedure was also used to check other dietary and maternal factors such as smoking, snuff use, 24-h and habitual intake of iodine, and maternal education. These were not significant predictors of HMIC and were therefore not included in the final regression model. All variables that were associated with HMIC (dependent variable) in a crude regression model (*p* < 0.10) were included in a preliminary multiple regression model. Excluded variables were reintroduced and those that were still significantly associated in this model (*p* < 0.10) were retained in the final model.

## 3. Results

### 3.1. Participant Characteristics

In this study, 42% of the mothers (*n* = 133) were exclusively breastfeeding, 35% were partially breastfeeding and 22% had ceased breastfeeding their infants. Only 23% reported use of supplements containing iodine the past 24-h. The majority (77%) of the mothers reported to never have smoked, while 20% reported to have smoked before, and 2% reported to be daily smokers. In this study, 4% of the mothers reported to be either vegetarians (omitting meat and fish, but including milk/and or eggs) (2%) or vegans (no animal products) (2%) (Table 1). 

The median (P25, P75) HMIC was 71 µg/L (45, 127), with a range between 11 and 280 µg/L. In total, 66% of the mothers had HMIC lower than 100 µg/L, 28% had sufficient HMIC between 100 and 199 µg/L, and 7% had HMIC above 200 µg/L (Figure 2).

The association between HMIC and infant’s age in weeks indicated a weak negative correlation, r_s_: −0.21 (*p* = 0.034). HMIC according to infant’s age in months is presented as a boxplot in Figure 3.

The median (P25, P75) UIC of the lactating mothers (*n* = 102) was 80 µg/L (52, 141), while it was higher (median 115 µg/L (55, 270) in those with weaned infants (*n* = 30), p-value for difference 0.020. The correlation between HMIC and UIC of those lactating was moderate, r_s_ = 0.33 (*p* = 0.001).

### 3.2. Calculated Habitual and 24-h Iodine Intake

The distribution of habitual and 24-h iodine intake for all mothers (*n* = 133) is presented in Table 2. The correlation between the lactating mother’s total habitual iodine intake (food and supplements) and total 24-h iodine intake (food and supplements) was strong, r_s_ = 0.62 (*p* = <0.001) (*n* = 102). The median (P25, P75) iodine intake from supplements was 150 µg/day by 24-h iodine intake (150, 150) (*n* = 30). By habitual intake, the median iodine intake (P25, P75) was also 150 µg/day (150, 200) (*n* = 30). 

Nearly all of the women in the postpartum-period had both total habitual (90%) and total 24-h iodine intake (94%) below the NNR of 200 µg/day. An even higher proportion did not reach the WHO recommendation of 250 µg/day—only 2% reached the WHO recommendation for maternal iodine intake by habitual intake and 3% by 24-h intake.

### 3.3. Iodine Knowledge Scores and Association with HMIC and Iodine Intake 

A high percentage, 81%, of the mothers reported knowing what iodine is. Approximately 25% reported to have heard about iodine but did not remember what it was. Only 16% reported to have received information from health care professionals about iodine, and 23% did not remember whether they had received any information. 

The mean ± SD total iodine knowledge score was 16 ± 6 points, and the median (P25, P75) score was 17 (14, 20) points. On a scale from 2 to 26 points, 8% had a poor knowledge score (< 6 points) and 9% had a low knowledge score (6–11 points). Half of the mothers (54%) had a medium knowledge score (12–19 points) and 29% had a higher knowledge score (20–26 points).

The total knowledge score among the lactating mothers (*n* = 102) correlated weakly with their HMIC, r_p_ = 0.20 (*p* = 0.041), their 24-h iodine intake (*n* = 132) (r_s =_ 0.24, *p* = 0.007), and their habitual iodine intake (*n* = 133) (r_s_ = 0.24, *p* = 0.006). The knowledge score did not correlate with UIC in all women (*n* = 131), r_s_ = 0.13 (*p* = 0.149), but was borderline correlated with UIC among the lactating women (*n* = 101) (r_s_ = 0.182, *p* = 0.068).

### 3.4. Predictors of Human Milk iodine concentration

In multiple linear regression models with HMIC as the dependent variable (Table 3), the mothers UIC µg/L, infant’s breastfeeding status, infant’s age in weeks, and quartile group 4 (high knowledge) compared to quartile 1 (poor knowledge) of total knowledge score were significant predictor of HMIC, explaining 36% of the variance in the HMIC (*p* ≤ 0.001). 

## 4. Discussion

This study is, to our knowledge, the first to present data on HMIC and iodine intake among lactating mothers in the inland area of Norway. We found that the women were mild to moderately iodine deficient, based on suboptimal iodine concentrations in human milk as well as in urine, and inadequate dietary iodine intake according to both habitual and 24-h food and supplement intake. We also found that the majority of the women had medium iodine knowledge. 

The median HMIC was below the suggested cut-off value at 100 µg/L. However, there is a lack of scientific consensus on sufficient concentration of iodine in human milk required to achieve iodine equilibrium in the infants [2,30,31,32]. Furthermore, HMIC may range from below 50 µg/L in iodine-deficient areas up to 100–200 µg/L in areas with iodine sufficiency [2,32]. Recently, Dold and co-workers suggested an estimated average requirement of 72 µg/day for 5-month-old infants [30]. Their recommendation corresponds to a HMIC > 92 µg/L, assuming a human milk consumption of 0.78 L/day [33]. The median HMIC of 71 µg/L in our study falls below this suggested cut-off value, and exclusively breastfed infants would receive approximately 55 µg/iodine per day based on a consumption of 0.78 L/day. Recommended daily intake (RDI) of iodine by the WHO for infants younger than 2 years is 90 µg/day [29,34]. However, there is a discrepancy between the intake recommendation of 90 µg/day and a median UIC cut-off value at 100 µg/L. Evaluation of iodine status is challenging in this age group. The WHO cut-off value for median UIC at 100 µg/L assumes a mean daily urine volume of 500 mL in infants and 92% iodine bioavailability [33], which corresponds to a mean daily intake of at least 55 µg [30,33]. Several studies have pointed out the need for more extensive research to define an estimated average requirement in infancy [30,35,36,37]. 

Furthermore, Dror and colleagues highlight the need for more research to establish an optimal HMIC in well-nourished women without risking excessive iodine intakes in their infants [38]. In our study, the mothers’ median UIC was lower than the WHO’s cut-off value at 100 µg/L for lactating women [28]. However, Dold and co-workers have suggested that UIC may not be an accurate measure of the iodine intake during lactation, due to increased fractional iodine excretion into the human milk at lower daily iodine intakes [3]. 

Overall, there is limited information on HMIC in the Nordic countries [10]. Only one study has previously assessed HMIC in Norwegian lactating women, and the results were in line with our findings of inadequate iodine status [25]. In Denmark, lactating women had an overall median HMIC of 83 µg/L, with supplement users (containing iodine) having higher (112 µg/L) HMIC than non-users (72 µg/L) [39]. Suggested insufficient HMIC has also been reported in Sweden (92 µg/L) [2] and in Finland (53 µg/L) [10]. Portugal is also a country that has previously been considered iodine sufficient. However, studies have found that pregnant and lactating women have inadequate iodine intake and insufficient iodine status, which emphasizes the need for monitoring iodine intake and status in countries with absent data [40]. 

There has been a decline in the consumption of milk, yoghurt and lean fish in Norway over the past decade [18,41]. Due to few sources of iodine in the Norwegian diet, the iodine intake has been highly dependent on preference for iodine rich foods (dairy products and seafood) and use of iodine-containing supplements, which may help explain the re-emergence of iodine deficiency. In 2016, the National Nutrition council reported that inadequate iodine intake was present in large subgroups of the population, especially among women of childbearing age. The council concluded that effective measures to secure adequate iodine intake should be undertaken immediately [42]. Norway has no regular monitoring of the iodine intake, but several studies published after the report from the Nutrition council have consistently shown inadequate iodine intake and insufficient iodine status in women of childbearing age [16,25,26,43,44,45,46,47]. Furthermore, a recent summary of iodine status in Norway concluded that insufficient iodine intake was prevalent, not only in women of childbearing age, but was also seen in other population groups, i.e., exclusively breastfed infants, the elderly, vegans and in a group of immigrants [11]. This study further supports these findings, with inadequate iodine intake among women of childbearing age. Despite the low iodine intake reported among these women, 41% of them reported to be planning a new pregnancy and 2% reported to be pregnant now. The iodine requirements increase during pregnancy and lactation, and therefore they are at an increased risk of hypothyroidism if the iodine intake is not increased during this period [17]. Starting a new pregnancy with low iodine status or having low iodine status during lactation may have consequences as thyroid disturbance in mothers and reduced neurodevelopment in children [6]. 

The lactating women in our study had medium iodine knowledge. This is somewhat better than previous studies reporting poor iodine knowledge among young women, pregnant and lactating women in Norway [26,27]. Other countries have also found poor iodine knowledge in this group [48,49,50]. Many of the mothers in our study wrongly identified the reason for why iodine is important. Having higher total knowledge score in our study was positively associated with increased HMIC. Therefore, increased iodine knowledge might be of importance for securing sufficient HMIC. 

HMIC has previously been seen to decline by time after delivery [25,51,52]. We also found that infant age was inversely associated with HMIC in the current study. The Norwegian Directorate of Health issued an advice for pregnant and lactating women in 2018, encouraging all women with low milk intake (less than two glasses per day) to take a daily iodine-containing supplement of 150 µg [41]. In this study, only 23% of the mothers reported use of iodine-containing supplement by 24-h intake—of which, 83% were still lactating. Studies have shown that supplementation with 150–200 µg iodine daily during lactation increases HMIC [2,39]. Also, Henjum et al. found that iodine-containing supplements contributed to the prediction of HMIC in a Norwegian setting [25]. Andersen et al. found that even if mothers used iodine-containing supplements, the Danish women did not reach the WHO’s RDI for iodine during lactation [39]. Leung and colleagues suggest that HMIC should be interpreted in relation to recent iodine intake, as they showed that oral ingestion of 600 µg potassium iodine (456 µg iodine) resulted in increased HMIC after an overnight fast [53]. Concerning the impact of recent maternal iodine intake on HMIC, maternal supplement use was not associated with HMIC in our study, perhaps due to sample size being too small. 

Lactating women who smoke have been found to have lower HMIC and lower UIC in their infants compared to non-smoking mothers [54]. Laurberg et al. suggested that this could be explained by thiocyanate, a goitrogenic substance present in the cigarette smoke, which inhibits iodine uptake in the mammary gland. In our study, smoking was not associated with HMIC, possible due to the sample size being too small. However, Henjum et al. found a significant negative association of smoking on HMIC, despite a low number of smokers (3%) [25]. Being a vegan and/or vegetarian has previously been associated with lower iodine intake than recommended and lower UIC, especially for vegans [47]. In our study, adherence to a vegetarian or a vegan diet was not associated with low HMIC µg/L, but the number of vegans and vegetarians were very small. 

A major strength of this study was using electronic questionnaires to obtain information that did not allow the respondents to leave blank answers. The detailed dietary assessment provided a detailed and accurate estimate of the iodine intake for each individual. This is a better measure of the iodine intake at the individual level than spot UIC, used in surveys of iodine status in populations. Compared to earlier Norwegian studies, we calculate the iodine content of all reported types of food in the FFQ, due to a recent update of iodine values in all foods in the food composition table [23]. Another strength is the assessment of iodine in both human milk and in urine samples. We collected two human milk specimens per women, and the specimens were pooled. As recent maternal iodine intake may influence HMIC and fluctuate throughout the day, two specimens per women give a more accurate level of the iodine concentration in human milk. The women were given equal instruction for obtaining the human milk specimens and to record timing for consumption of iodine-containing supplements. 

This study also has limitations. We used convenience sampling to recruit participants, and thus the participants were not a representative sample of the total population of lactating women. In lactating women, a particular challenge pertains to what is denoted as “dazed due to breastfeeding”. Some of the mothers in this study reported alarmingly low food intake in general. This may be due to lack of time and stress during the day. Hence, the possibility of underreported food intake by the mothers may be particularly relevant in this study population. 

## 5. Conclusions

The lactating women in this study were mild to moderately iodine deficient. The majority had insufficient HMIC and UIC, and inadequate iodine intake calculated from food and iodine-containing supplements. In addition, they had medium knowledge about iodine. Targeting iodine knowledge among women of childbearing age may be important to secure sufficient iodine status in Norwegian breastfed infants. 

## Figures and Tables

**Figure 1 nutrients-12-00630-f001:**
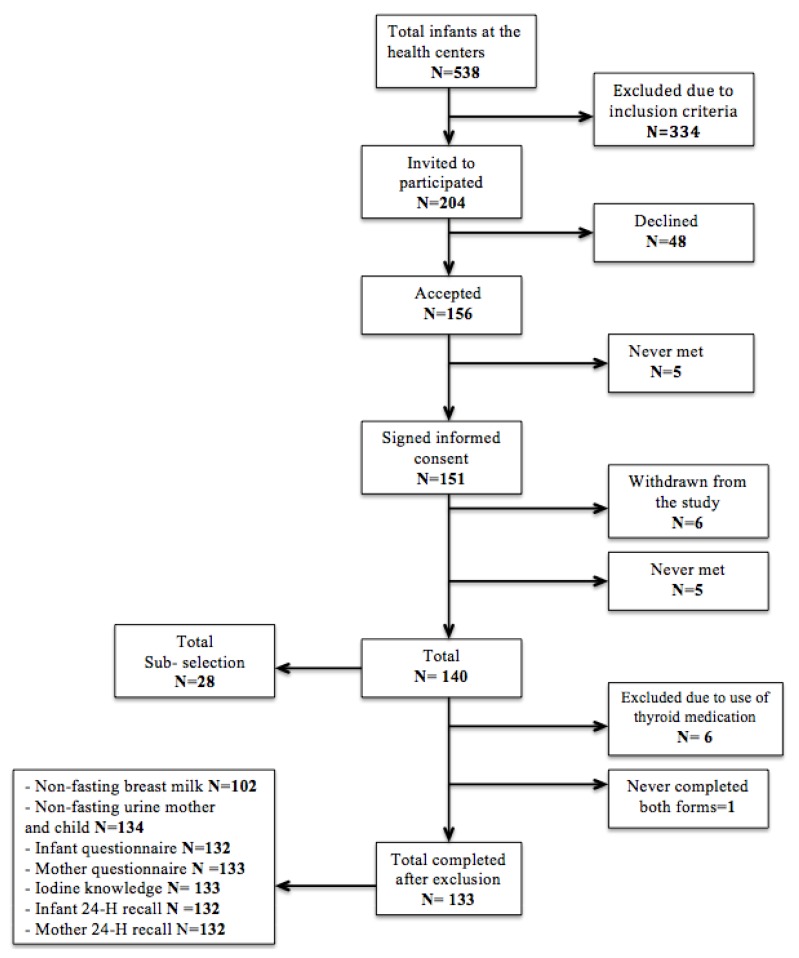
Flow chart of the recruitment and sample selection. In this article, only data regarding the mothers are presented.

**Figure 2 nutrients-12-00630-f002:**
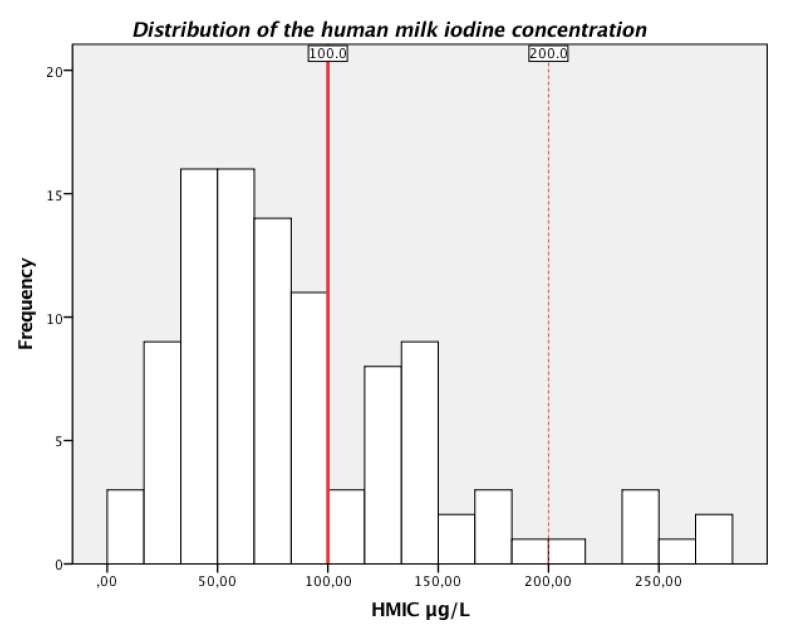
Frequency distribution of human milk iodine concentration (HMIC µg/L) among the lactating mothers (*n* = 102). The red solid line shows the suggested cut-off value for sufficient HMIC ≥100 µg/L, and the range between the red solid and red dotted line shows the suggested range of sufficient HMIC (100–199 µg/L).

**Figure 3 nutrients-12-00630-f003:**
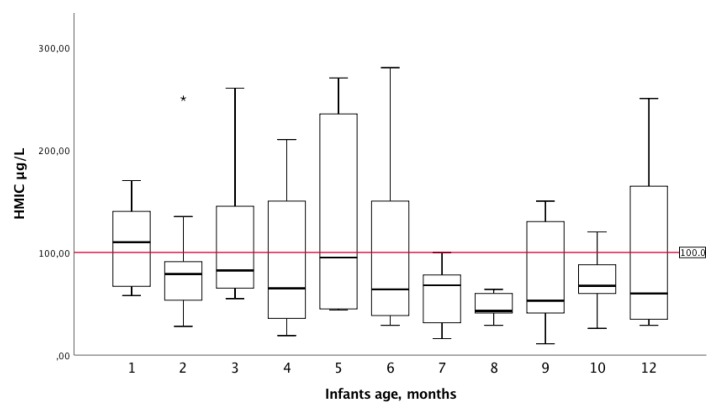
Human milk iodine concentration (HMIC µg/L) by infant’s age (months). Box plot details: The horizontal line in each box indicates the median HMIC; the box indicates the interquartile range (IQR) (25th percentile to 75th percentile); the whiskers represent observations within 1.5 times the IQR; and the stars indicate observations more than 1.5 times the IQR away from the box, considered as outliers. The horizontal red line represents the cut-off value considered to provide adequate iodine in infants. Infants still breastfed at 13 and 14 months (recruited at 12 months of age) were excluded in this figure, due to low breastfeeding frequency and few participants.

**Table 1 nutrients-12-00630-t001:** Characteristics of the mothers and infants (*n* = 133).

Characteristic		N (%)
Maternal age	Mean (SD), min–max	31.4 (4.6) 20–42
Marital status	Single	1 (1)
	Cohabiting	82 (61.2)
	Married	48 (35.8)
	In a relationship	2 (2)
Country of origin	Norway	120 (90)
	Other (11 different countries)	13 (10)
Pre pregnancy, BMI, kg/m^2^	< 18.5	4 (3)
	18.5–24.9	93 (70)
	25–30	21 (16)
	> 30	15 (11)
BMI by inclusion, kg/m^2^	< 18.5	4 (3)
	18.5–24.9	83 (62)
	25–30	30 (23)
	> 30	16 (12)
Planning new pregnancy	Yes	54 (41)
Pregnant now	Yes	2 (2)
Infants age, in months ^1^	1–4	54 (41)
	5–7 months	27 (20)
	8–11 months	35 (26)
	12–13 months	17 (13)
Educational level	<12 years	6 (5)
	12 years	19 (15)
	1–4 years college/university	52 (39)
	>4 years college/university	56 (42)
Smoking status	Never smoked	103 (77)
	Smoked before pregnancy	27 (20)
	Occasionally during pregnancy	0
	Daily smoker during pregnancy ^2^	3 (2)
Dietary supplement use	Any supplement (FFQ)	76 (57)
	Iodine-containing supplement past 24-h	30 (23)
Adhering to a vegetarian or vegan diet		6 (4)
Diet includes dairy	No	9 (7)
	Occasionally	22 (16)
	Yes, daily	103 (77)
Self-reported hypothyroidism, no use of thyroid medication		2 (2)

^1^ Seven children in the 13 months category were 13 months, plus 1 or 2 weeks (recruited at age 12 months); ^2^ number of cigarettes daily: 12 ± 3 during the whole gestation. BMI, Body Mass Index.3.2. Iodine Concentrations in Human Milk (HMIC) and Urine (UIC).

**Table 2 nutrients-12-00630-t002:** Calculated habitual and 24-h iodine intake (µg/day) for all mothers (*n* = 133).

Iodine Intakes	Median	P25 ^1^	P75 ^2^	Mean ± SD
Habitual iodine intake from food, µg/day	35	20	52	40 ± 26
Total habitual iodine intake, µg /day	42	28	86	75 ± 73
The 24-h iodine intake from iodine rich foods only, µg /day	23	15	38	42 ± 44
Total 24-h iodine intake, µg /day	32	16	160	78 ± 79

^1^ P25 = 25th percentile; ^2^ P75 = 75th percentile.

**Table 3 nutrients-12-00630-t003:** Predictors of human milk iodine concentration (HMIC) in the lactating mothers (*n* = 100).

Dependent Variable	Predictor Variables	Unstandardized Beta	*P*	Standardized Beta	95% CI Beta
HMIC µg/L,	Constant				
	UIC µg/L ^1^	0.26	< 0.001	0.42	(0.16, 0.37)
	Breastfeeding status ^2^	19.9	< 0.001	0.35	(10.5, 29.3)
	Infant age, weeks	−1.17	0.001	−0.31	(−1.82, −0.52)
	Knowledge score ^3^	9.63	0.035	0.18	(0.70, 18.56)

^1^ UIC µg/L excreted in the same 24-h as HMIC; ^2^ infants breastfeeding status (exclusively breastfed compared to partially breastfed); ^3^ quartile group of total knowledge of iodine (Q4 compared to Q1).

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
