# Peer review of "Mild to Moderate Iodine Deficiency and Inadequate Iodine Intake in Lactating Women in the Inland Area of Norway"

_nutrients, 2020, doi:10.3390/nu12030630_

Round 1

Reviewer 1 Report

The authors have done a great job presenting their work.  A difficult task with many variables to deal with.  A word of caution about the analytical method for iodide (mass spec).  The US Agency, CDC, found that frozen iodine samples of breast milk degraded with time (work unpublished).  The belief is that iodide was converted to protein bound forms and thus not measured by the mass spec. Using old analytical methods to measure total iodine may be important to ensure the proper measurements in breast milk.

An expanded discussion of previous iodine deficiency studies for lactating mothers would improve the caliber of this paper (e.g.,  Costeira et al. 2009, 2010, Mulrine et al. 2010), which includes iodide levels in maternal and infant urine and breast milk and in some cases serum thyroid hormone levels. These papers and others are referenced by Fisher et al. 2016 PLoS ONE11(3).

An expanded discussion about iodine requirements in neonates, term and preterm, and its uncertainty would for the first months of life would improve the quality of this paper.  

Author Response

Point 1: ”A word of caution about the analytical method for iodine (mass spec). The US Agency, CDC found frozen iodine samples of breast milk degraded with time (work unpublished). The belief is that iodide was converted to protein bound forms and thus not measured by the mass spec. Using old analytical methods to measure total iodine may be important to ensure the proper measurements in breast milk.

Response 1: The method validation showed no sign of loss of analyte. Concurrent analysis of SRM confirm method trueness, however, method precision also contribute to measurement uncertainty. The reproducibility is determined on replicates of the frozen samples, not on CRM. It is difficult to comment on the observation you are referring to without knowledge on the sample preparation used. Choice of reagents used in the sample preparation is important.

Point 2: “An expanded discussion of previous iodine deficiency studies for lactating mothers would improve the caliber of this paper (e.g., Costeira et al. 2009, 2010; Mulrine et al. 2010), which includes iodine levels in maternal and infant urine and breast milk and in some cases serum thyroid hormones levels. These papers and others are referenced by Fisher et al. 2016 PLoS ONE11(3). “

Response 2: We agree that it is important to discuss the previous studies in other countries concerning lactating mothers. We have now added a sentence about the iodine status of lactating women in Portugal, with the recommended references Costeira et al. 2009 (see lines 294-297, page 9 and 10) However, we already refer to the suggested reference Mulrine et al. 2010 (see line 325, page 10), by mentioning that a decline in the breast milk by the time after delivery has previously been seen.

Point 3: “An expanded discussion about iodine requirements in neonates, term and preterm, and its uncertainty would for the first months of life improve the quality of this paper.”

Response 3: Thank you for this comment. We have now added a short elaboration of the discrepancy between the intake recommendation and cut-off recommendation by WHO at this stage of age (see lines 277-282, page 9).

Reviewer 2 Report

Thank you for providing an opportunity to review this paper. 

The authors were interested in assessing iodine intake and status in lactating women from Norway. Overall, the information is well presented and clear, just some minor comments:

  • Throughout the manuscript, the term 'human milk' would be more accurate.
  • Introduction: It would be nice to see the breastfeeding rate (%) of Norway. Also, I would include a couple of sentences to highlight the importance/function of iodine in infant development to really engage the audience in the reason why this is important to research.
  • M&M-Collection of breast milk and urine samples: More information on the collection of breast milk is needed.
    • Were women fasting for the first collection? How would fasting/no fasting affect iodine concentrations?
    • During what lactation stage were samples collected? Did you have a criteria for this? It is not clear if you collected the samples within the same lactation period for all the mothers.
    • Would time of the day in which the sample was collected affect the iodine concentrations? Was this the reason why the two samples were pooled?
  • Table 1: Specify to which point 'current' BMI is referring to.
  • Table 2: Would it be possible to include how much iodine intake women were getting from supplements?
  • Conclusion: I would argue that iodine knowledge would also be important pre-pregnancy and during pregnancy.

Author Response

Point 1: ”Throughout the manuscript, the term `human milk` would be more accurate.”

Response 1: Thank you for this comment. We agree that the term `human milk` are more accurate and have now changed this throughout the manuscript.

Point 2: Introduction: It would be nice to see the breastfeeding rate (%) of Norway. Also, I would include a couple of sentences to highlight the importance/function of iodine in infant development to really engage the audience in the reason why this is important to research.

Response 2: We agree and have added information of breastfeeding rate (%) in Norway by age based on the national breastfeeding report (see lines 55-57 page 2).

Concerning your comment to include a couple of sentences to highlight the importance/function of iodine in infant development to engage the audience, we believe that this is being described initially (see lines 43-49, page 2).

However, we added a sentence describing the possible consequences of inadequate thyroid hormone production in infancy (see lines 45-46, page 2) in addition to what was already explained. We also added the estimated iodine deficiency prevalence among infants in Europe from the Krakow declaration in 2018 (see lines 49-51, page 2). We hope this change suits you well.

Point 3: Were women fasting for the first collection? How would fasting/no fasting affect iodine concentrations?

Response 3: Urinary iodine concentration follows a circadian rhythm (Als et al, see https://www.ncbi.nlm.nih.gov/pubmed/10770167): a study with 3023 spot urine samples in adults and children. Both of the breast milk samples and the urine sample was instructed to be non-fasting. We have now further specified that both were non-fasting samples (see lines 99-101, figure 1, page 3). We did not use fasting samples, and WHO do not recommend fasting samples (WHO, 2007).

Both human milk and the urinary iodine concentration can be highly inflicted by recent iodine intake and fluid intake at individual level, however variation in hydration among individuals generally evens out in a population, and median human milk and urinary concentration is recommended to use. 

Point 4: During what lactation stage were samples collected? Did you have a criteria for this? It is not clear if you collected the samples within the same lactation period for all the mothers.

Response 4: According to the inclusion all healthy mothers who could read and write Norwegian, and had a healthy infant between 0-12 months at study start were invited to participate (see lines 87-89, page 2). This study included those lactating and those not, lactation was not a criteria in this study. However, this means that the human milk specimens are provided from 0-12 months and the lactation stage depends on the age of the infant. We hope that this clarified your question.

Point 5: Would time of the day in which the sample was collected affect the iodine concentrations? Was this the reason why the two samples were pooled?

 Response 5: Recent maternal iodine intake may influence the human milk iodine concentration, which fluctuates throughout the day; thus two specimens will give a more accurate output level. Furthermore, iodine concentration has been found highest in colostrum and declining gradually as lactation progresses, we collected two human milk specimens per women, one in the morning and one in the afternoon; one with fore-milk and one with hind milk to provide a more precise measurement of the daily iodine output in the human milk.

Point 6: Table 1: Specify to which point 'current' BMI is referring to.

Response 6: Thank you for this comment. We changed current BMI to “BMI by inclusion” to avoid misunderstanding (see table 1, page 6).

Point 7:Table 2: Would it be possible to include how much iodine intake women were getting from supplements?

Response 7:  Thank you for this comment, we have now included a sentence about how much iodine the women were getting from supplements (see lines 234-235, page 82).

Point 8: Conclusion: I would argue that iodine knowledge would also be important pre-pregnancy and during pregnancy.

Response 8: Thank you for this comment. We agree that iodine knowledge will be important pre-pregnancy and during pregnancy as early pregnancy is a particularly crucial period for fetal brain development.  We have now edited the conclusion to apply to all women of childbearing age (see line 377, page 14).

This manuscript is a resubmission of an earlier submission. The following is a list of the peer review reports and author responses from that submission.